# Research on Particle Swarm Compensation Method for Subdivision Error Optimization of Photoelectric Encoder Based on Parallel Iteration

**DOI:** 10.3390/s22124456

**Published:** 2022-06-12

**Authors:** Han Hou, Guohua Cao, Hongchang Ding, Kun Li

**Affiliations:** 1Mechanical Engineering Faculty, Changchun University of Science and Technology, Changchun 130022, China; eric_houyz@163.com (H.H.); dinghc@cust.edu.cn (H.D.); hgclikun@163.com (K.L.); 2Chongqing Research Institute, Changchun University of Science and Technology, No. 618 Liangjiang Avenue, Chongqing 401135, China

**Keywords:** interpolation error compensation, grating moiré fringe, FPGA, particle swarm optimization algorithm

## Abstract

Photoelectric encoders are widely used in high-precision measurement fields such as industry and aerospace because of their high precision and reliability. In order to improve the subdivision accuracy of moiré grating signals, a particle swarm optimization compensation model for grating the subdivision error of a photoelectric encoder based on parallel iteration is proposed. In the paper, an adaptive subdivision method of a particle swarm search domain based on the honeycomb structure is proposed, and a raster signal subdivision error compensation model based on the multi-swarm particle swarm optimization algorithm based on parallel iteration is established. The optimization algorithm can effectively improve the convergence speed and system accuracy of traditional particle swarm optimization. Finally, according to the subdivision error compensation algorithm, the subdivision error of the grating system caused by the sinusoidal error in the system is quickly corrected by taking advantage of the high-speed parallel processing of the FPGA pipeline architecture. The design experiment uses a 25-bit photoelectric encoder to verify the subdivision error algorithm. The experimental results show that the actual dynamic subdivision error can be reduced to ½ before compensation, and the static subdivision error can be reduced from 1.264″ to 0.487″ before detection.

## 1. Introduction

As a precise position-sensing device, the photoelectric encoder has been widely used in industrial manufacturing and assembly processes [1,2,3]. At present, the precision measurement field puts forward higher requirements on the precision and resolution of photoelectric encoders, and it is difficult to meet the actual requirements only by improving the processing accuracy of the grating. Therefore, it is of great significance to study the correction method of the subdivision error to improve the angle measurement accuracy of photoelectric shaft angle encoders [4,5,6,7].

The grating moiré signal usually has the following typical errors: DC error offset, equal amplitude error, quadrature error, sinusoidal deviation [8], and harmonic error. The above error factors are easily disturbed by the external environment during the experiment and are directly reflected in the signal as the change in the subdivision error. In order to improve the angular resolution of moiré gratings, the grating moiré signals generated by the moiré grating are usually processed by means of subdivision [9].

At present, a grating moiré signal compensation model is established for the subdivision error existing in the output signal [10], and then the waveform parameters of the signal are solved by the error fitting algorithm. The method of further realizing grating moiré signal subdivision compensation is relatively common [11,12,13]. Sun proposed a triangular signal integral compensation method for the subdivision error of the moiré triangle [14,15]. During the experimental test, the error compensation system reduced the subdivision error to ⅓ of the original error. The South Korean company LG uses the grating movement measurement method to compensate for the phase subdivision error in the signal acquisition process [16]. In 2017, Zhang et al. of China Jiliang University proposed a digital quadrature deviation real-time compensation method based on the CORDIC algorithm [17]. This method solves the problem that the principle error of quadrature error compensation in the previous CORDIC algorithm and the angle calculation sensitivity of the signal interval are low. Real-time compensation for the quadrature deviation is realized, but the CORDIC algorithm is used many times in the process of angle calculation and compensation, and the system hardware resources are occupied and the system delay is more serious.

To sum up, the above encoder subdivision error compensation methods have the following problems: First, most subdivision error compensation methods cannot achieve comprehensive subdivision error compensation, and only analyze and correct the subdivision error under ideal conditions; second, the methods used in the above error compensation methods are all software-based subdivision error compensation methods. Post-processing of error data can only satisfy the compensation of subdivision errors under laboratory conditions, and cannot meet the real-time error compensation effect.

This paper takes the arctangent principle as the theoretical basis of error compensation, analyzes the mathematical model of the photoelectric signal error model, and proposes a parallel iterative multi-group optimization search algorithm. A parallel iterative multi-group optimization search algorithm based on the von Neumann topology model is designed to solve the optimal value of the nonlinear waveform equation obtained from the sampled data. The multi-region parallel search is realized by adaptive region division of the search target region, and the key optimal solution of the algorithm is obtained more efficiently and accurately. The principle of information sharing between subgroups can make it easier for particle swarm optimization to get rid of the dilemma of falling into local optimum in complex high-latitude optimization problems, and effectively improve the global search ability and comprehensive performance of particle swarm optimization.

In this paper, the 25-bit photoelectric encoder is used as the experimental object to verify the effect of the algorithm. The implementation results show that the multi-swarm optimization search algorithm proposed in this paper can effectively improve the convergence speed of the particle swarm algorithm and the accuracy of the optimal value, and effectively improve the calculation speed and system accuracy of the error compensation algorithm. The subdivision error existing in the output signal of the encoder can be effectively compensated for, and the problem of system instability caused by the change in signal quality in a harsh environment can be effectively improved. The dynamic subdivision error after compensation can be reduced to that before compensation, and the static subdivision error can be reduced from 1.264″ before detection to 0.487″, and the compensation effect is better.

## 2. Analysis of Subdivision Error

As shown in Figure 1, the main working principle of the photoelectric encoder is photoelectric conversion, but its output is digital. There are several concentric code tracks on the code disc of the photoelectric encoder. Each code track is formed by intersecting light-transmitting and non-light-transmitting sector intervals. The number of code tracks is the number of binary digits of the code disc where it is located. The two sides of the code wheel are the light source and the photosensitive element, respectively. During the operation of the system, the LED light source emits monochromatic light, the scale grating in the grating pair rotates with the rotating shaft, and the indicating grating is fixed on the body shell. The outgoing light is received by the photodetection device through the sub-slits of the grating. The collected grating signal can be converted into the digital signal of the system by the photoelectric detection device. This article takes the 25-bit optical encoder produced by Yuheng Optical Co., Ltd. (Changchun, China), as the test object. This model is a hollow shaft absolute rotary encoder, the size of the body shell of this type of product is 90 mm, and the system resolution is $2^{25}$.

The grating moiré signal is affected by sinusoids and is usually a periodic signal between a triangular wave and a sine wave, and the sine wave is often used as a model in the calculation process. The main reason for the subdivision error is that the photoelectric signal of moiré grating fringes deviates from the standard of sine and cosine signals. The ideal moiré fringe photoelectric signal can be expressed as (1): (1)$$\left\{ \begin{array}{l} u_{1}=A\sin\theta \\ u_{2}=B\cos\theta \end{array}\right.$$

Affected by the scribing accuracy of the code disc, the accuracy of the shaft system, and the quality of the photoelectric signal, the complete expression of the precise code signal can be expressed as:(2)$$\left\{ \begin{array}{l} U_{1}=a_{0}+a_{1}\sin\theta +\sum_{i=2}^{19}a_{i}\sin(i\theta +\varphi_{i1})+\delta_{e}\\ U_{2}=b_{0}+b_{1}\cos\theta +\sum_{i=2}^{19}b_{i}\cos(i\theta +\varphi_{i2})+\delta_{e} \end{array}\right.$$

Among them, $a_{0},b_{0}$ represents the DC component of the signal, which causes the DC level drift; $a_{m},b_{m}$ represents the fundamental wave of the signal amplitude, which causes the amplitude subdivision error; and $\sum_{i=2}^{19}a_{i}\sin(i\theta +\varphi_{i1}),\sum_{i=2}^{19}b_{i}\sin(i\theta +\varphi_{i2})$ is the higher harmonic error, which causes the harmonic subdivision error. In the spectrum analysis of higher harmonic errors later, the influence of harmonic errors higher than the 19th order on the signal can be ignored, so in this paper, we select the 19th order harmonic error as the highest; $\varphi_{i1},\varphi_{i2}$ is the cause of the phase error. In the actual measurement process, there will also be electrical noise interference, which can be represented by $\delta_{e}$.

From the above expression, we can conclude that the subdivision angle can be calculated by the arctangent function [18] formula:(3)$$\theta =\arctan\frac{U_{1}}{U_{2}}$$

When $u_{1},u_{2}$ is in an ideal state, θ in the sine–cosine waveform signal is the ideal value of the subdivision angle. The angle value $\theta_{r}=\theta +\Delta \theta =\arctan\frac{U_{1}}{U_{2}}$ in the actual subdivision process, where Δθ is the subdivision error.

Because the data output of the moiré grating adopts a digital encoding method, quantization errors will inevitably occur. When the encoder is located at a certain angle θ, the theoretical output voltage values of the encoder’s moiré fringe are sinθ and cosθ, but the data values generated in the actual measurement process are $U_{1}$ and $U_{2}$, resulting in a certain interpolation error.

### 2.1. Sine Error Analysis

When the grating signal only has the amplitude error component, the amplitude error component Δr is introduced into the expression of the interpolation error. In the Lissajous graph, the amplitude difference between the X-axis and the Y-axis is the ellipse state.

As shown in Figure 2, when the actual position of the detection signal is point B, the amplitude of the sine is r and the amplitude of the cosine is r−Δr. At this time, the amount of error of the cosine amplitude in the Y-axis direction is l.
(4)$$l=\Delta r\sin\theta_{r}$$
(5)$$r\sin(\Delta \theta )=l\sin(\frac{\pi }{2}+\theta_{r}-\Delta \theta )$$
(6)$$r\sin(\Delta \theta )=\Delta r\sin\theta_{r}\cdot \cos(\theta_{r}-\Delta \theta )$$
(7)$$\Delta \theta =\theta_{r}-\arctan\frac{(r-\Delta r)\sin\theta_{r}}{r\cos\theta_{r}}$$
(8)$$\tan\Delta \theta =\frac{\tan\theta_{r}-\frac{(r-\Delta r)\sin\theta_{r}}{r\cos\theta_{r}}}{1+\tan\theta_{r}\frac{(r-\Delta r)\sin\theta_{r}}{r\cos\theta_{r}}}$$

After simplification, the error expression of Δθ can be obtained:(9)$$\Delta \theta =\arctan\frac{\Delta r\sin(2\theta +2\Delta \theta )}{\cos(2\theta +2\Delta \theta )\Delta r+2r-\Delta r}$$

Let $K=\frac{2r}{\Delta r}-1$, $\theta_{k}=2\theta_{r}$ and derivate the above formula:(10)$$(\Delta \theta )\prime =[\frac{1}{1+{\left(\frac{\sin\theta_{k}}{K+\cos\theta_{k}}\right)}^{2}}](\frac{\sin\theta_{k}}{K+\cos\theta_{k}})\prime$$
(11)$$(\Delta \theta )\prime =\frac{1+K\cos\theta_{k}}{1+2K\cos\theta_{k}+K^{2}}$$

In $\theta_{r}=n\pi /4$n=1,3,5,7, obtain the maximum error value $\Delta \theta_{\max}=\arctan\Delta r/(2r-\Delta r)$.

Obtain the minimum error at $\theta_{r}=n\pi /2$, $\Delta \theta_{\min}=0$.

Under normal circumstances, the amplitude of a single harmonic in the grating moiré signal should not exceed 20% of the fundamental amplitude. According to the arctangent principle, the higher the sine exponent, the more obvious the error trend caused by the change in the theoretical angle due to the interpolation error. When the sine index reaches the peak value of 0.2 within the range, the maximum interpolation error reaches 11.30″. Through MATLAB, we can simulate the relationship between the sine exponent and the angle error curve as shown in Figure 3.

### 2.2. Analysis of Higher Harmonic Error

In the actual signal measurement process, the high-precision waveform signal is close to the sine wave. When the fine code signal decreases, the subdivided waveform approaches the triangle wave. The error of higher harmonic components depends on the acquisition quality of the moiré fringe signal, and the processing algorithm of signal subdivision can be calculated by the θ=arctansinθ/cosθ method. The method of checking the subdivision table is often used in the algorithm program. When the degree of sine and triangle waveform of the subdivision signal waveform is low, a subdivision table error will occur. The error between the standard sine wave subdivision table and the triangle wave subdivision table is about 8.2%. When the sampled signal only contains high-order harmonic errors, Formula (2) can be rewritten as:(12)$$\left\{ \begin{array}{l} U_{1}=a_{1}\sin\theta +\sum_{i=2}^{5}a_{i}\sin(i\theta +\varphi_{i1})\\ U_{2}=b_{1}\cos\theta +\sum_{i=2}^{5}b_{i}\cos(i\theta +\varphi_{i2}) \end{array}\right.$$

From the previous derivation of the relationship between the actual angle value and the subdivision error value, $\theta_{r}=\theta +\Delta \theta$. Introduce Formula (12) into the algorithm:(13)$$\Delta \theta =\theta_{r}-\arctan\frac{a_{1}\sin\theta_{r}+\sum_{i=2}^{5}a_{i}\sin i\theta_{r}}{b_{1}\cos\theta_{r}+\sum_{i=2}^{5}b_{i}\cos i\theta_{r}}$$

Arranging Formula (13), we can get:(14)$$\tan\Delta \theta =\frac{\tan\theta_{r}-\frac{a_{1}\sin\theta_{r}+\sum_{i=2}^{5}a_{i}\sin i\theta_{r}}{b_{1}\cos\theta_{r}+\sum_{i=2}^{5}b_{i}\cos i\theta_{r}}}{1+\tan\theta_{r}\frac{a_{1}\sin\theta_{r}+\sum_{i=2}^{5}a_{i}\sin i\theta_{r}}{b_{1}\cos\theta_{r}+\sum_{i=2}^{5}b_{i}\cos i\theta_{r}}}$$

In order to further consider the influence of harmonic error on the output signal of the photoelectric encoder, Fourier harmonic analysis was carried out on the sampled signal. The analysis results are shown in Figure 4. It can be seen that the influence of harmonics above the 19th order is approximately negligible. The harmonic errors that can significantly affect the subdivision accuracy of the output signal of the photoelectric encoder are the second, third, and fifth harmonic errors. Therefore, it is further verified that in Formula (12), our error and upper limit for higher harmonics are the fifth harmonic.

## 3. PSO Algorithm Error Compensation

### 3.1. Principle of Error Correction

Usually in a grating optical system, the slit disk adopts a four-phase indicating grating, and the collected photoelectric moiré fringe signal will be split into four current signals that are out of phase with each other. After the current–voltage conversion process, the signal is amplified, and the A/D data conversion processor is used to convert the sampled signal to facilitate decoding and other tasks. The converted electrical signal can be amplified, filtered, and subdivided by the data processing system, and finally the output angle code can be obtained. Ideally, when the phase difference of the photoelectric signals output by the four groups is π/2, the harmonic error and the DC level component existing in the signal can be canceled by the method of differential complementarity. However, in the actual signal acquisition process, the phase of the acquired signal is difficult to ensure a standard phase difference, so the subdivision signal after the acquired signal is differentiated will contain different degrees of harmonic interference.

In the grating detection process, the quality of the fine code signal will directly determine the accuracy of the subdivision precision. In order to improve the resolution and subdivision accuracy of the encoder in a complex working environment, it is necessary to correct the sinusoidal deviations and harmonic errors contained in the sampled signal. In this paper, the non-uniform sampling method is used to simulate the actual working state, and the spectrum analysis of the moiré fringe signal through discrete Fourier transform can be obtained. The differential output signal of the grating moiré fringe photoelectric signal can be expressed as the mathematical analysis expression model shown in the previous Formula (2).

According to the Fourier space transform theory, it can be obtained that the periodic photoelectric moiré fringe signal actually output by the grating in the actual output process can be expressed by the Fourier series expansion. It can be seen from the previous harmonic error amplitude–frequency analysis that in the precision grating system, the harmonic components that have the greatest influence on the subdivision error of the photoelectric moiré signal are the second, third, and fifth harmonics of the photoelectric signal. In this paper, taking the third harmonic with the highest influence as an example, the third harmonic component in the signal is represented by *U*′, and the maximum harmonic subdivision error can be expressed as Formula (15):(15)$$\Delta \phi_{s}\approx \pm \arcsin U^{\prime }$$

When $U^{\prime }$ is close to zero, the maximum subdivision error can be approximately expressed as $\Delta \phi_{s}\approx \pm U^{\prime }$, which is converted to the relative value expression of the angle:(16)$$U^{\prime }=\Delta X_{s}/A(T/2\pi )$$

When the subdivision number *T* = 1024 and $\Delta X_{s}/A\leq 1$, the third harmonic is $U^{\prime }\leq 0.6\%$; if the subdivision number is halved, *T* = 512, and the relative value of the maximum subdivision angle is less than 1, the third harmonic is $U^{\prime }\leq 1.2\%$.

According to the actual output of the grating moiré fringe signal, the actual output is calibrated by means of parameter identification. Within the experimental value range, we can express the subdivision error compensation model as Formula (17):(17)$$\theta =\arctan(U_{1}/U_{2})-\Delta \theta$$

### 3.2. Moiré Fringe Interpolation Error Compensation Model

After discrete error acquisition and differential processing, a set of ideal sine and cosine moiré fringe signals can be obtained. We use the tangent method to construct the subdivision error compensation function:(18)$$f(\theta )=\frac{U_{1}}{U_{2}}=\frac{A\sin\theta }{B\cos\theta }$$

Because the signal is considered to be in an ideal state at this time, A=B. At this time, the rotation angle can be obtained by Formula (3).

However, in practice, the amplitude-to-phase relationship of the sine signal is nonlinear. We use U(θ) to represent the state of the sine and cosine signals in different intervals.
(19)$$U(\theta )=\left\{ \begin{array}{ll} \tan\theta =\frac{\left|A\sin\theta \right|}{\left|B\cos\theta \right|} & \left|A\sin\theta \right|\leq \left|B\cos\theta \right|\\ \cot\theta =\frac{\left|B\cos\theta \right|}{\left|A\sin\theta \right|} & \left|B\cos\theta \right|\leq \left|A\sin\theta \right| \end{array}\right.$$

Using the principle of the tangent method, at this time, the effective angle information can be obtained from the amplitude of the collected moiré fringe photoelectric signal. $\arctan(A_{1}\sin\theta /B_{1}\cos\theta )$ is a multivalued function in the [0,2π] phase angle range, which is converted into a numerical function and then operated on.

The tangent function and the cotangent function have the following relationship in numerical value:(20)θ=π2-arccot(Acosθ/Bsinθ)

A single acquisition period was selected as the test sample, and the acquisition signal was divided into eight parts $S_{1}\sim S_{8}$, with the intersection point and zero point of the sine and cosine moiré signals as the dividing line, and π8 used as a single group step size. In a single group of steps, the tangent function represents the single-angle value function about θ, and the subdivision output value is calculated and obtained according to the number of intervals. The relationship between the number of intervals and the subdivision value is shown in Figure 5.

It can be seen from the image properties of the tangent function that the function value of the tangent function at π/2 and 3π/2 tends to infinity, which will directly lead to the overflow of the angle calculation value. In order to reduce the influence of the above errors on the output results, a set of overdetermined nonlinear equations for waveform parameter identification are established for the collected periodic moiré fringe photoelectric signal data. The equation can obtain the optimal solution of the overdetermined nonlinear equation system and convert it into the identification result of the waveform parameters.

According to the subdivision error compensation model mentioned above, in this paper, we adopt the nonlinear least squares estimation method to deal with the optimal solution of parameters in the nonlinear equation of moiré fringe signal. The moiré fringe output signal can be obtained from Formula (2), and its mathematical model is recorded as:(21)$$u=f[(\theta_{1},\alpha_{1}),(\theta_{2},\alpha_{2}),\cdots ,(\theta_{i},\alpha_{i})]$$

u represents the moiré signal output value, and $\alpha_{i}$ represents the parameter value to be identified in the output waveform equation. In the actual error signal processing system, the error component mainly exists in the observation component u. If u is repeatedly output m times at the time $\left[t_{begin},t_{end}\right]$, the output value deviation of the moiré output signal observation function signal can be expressed as:(22)$$e_{i}=u_{i}-f(\alpha_{i})$$

The merit function solution model of the error estimation algorithm is established according to the principle of least squares. The magnitude deviation contained in the observation u will directly affect the stability of the cost function model. The signal within the output period of the grating is sampled, and the error squared sum e of the sampled signal is used as the cost function of the algorithm to estimate the parameters.
(23)$$e=\min\sum_{i=1}^{N}{\left[u_{i}-f(\alpha_{i})\right]}^{2}$$

In order to improve the robustness of the system and reduce the influence of random errors, the number of sample points should be increased during the acquisition period of the grating moiré fringe signal. An overdetermined nonlinear equation for parameter identification is established, and the optimal solution obtained is used as the parameter identification solution of the grating moiré waveform equation.

In order to improve the system accuracy of particle swarm optimization and the problem of low convergence efficiency, this paper is based on traditional particle swarm optimization. By dividing the target population search area by Latin hypercube sampling, parallel population iteration is performed on the particle individuals in the grouped area. The local optimal information is exchanged and compared using the von Neumann topology to obtain the optimal global target value. It effectively improves the search efficiency of population target particles and quickly filters out invalid search areas according to the system accuracy requirements, improving the search accuracy of target parameters.

### 3.3. Particle Swarm Optimization Compensation Principle

The particle swarm optimization method initially simulates the predation behavior of birds and uses the information interaction between population individuals to determine the optimal location in the population search area. It guides the individual particles in the whole group to iteratively converge to the optimal individual position while retaining the individual diversity characteristics.

The PSO algorithm can be described as a process in which n particles in an N-dimensional search space determine the global optimum through continuous iterative updates of the search speed and search position. In the startup phase of the algorithm, the particles in the search space are initialized, and a “position” is assigned to a single particle—that is, the potential solution of the desired target problem. The method for judging the feasibility of the current solution is to bring the individual particle $x_{i,d}$ into the objective function to calculate the fitness value of the calculator, and based on this, measure the pros and cons of the potential solution among all solutions. For individual particles, each iteration compares an individual optimal position with the current search velocity vector. By summarizing and comparing the states of all particles in the search space, the optimal value of the population will be obtained—that is, the optimal value of the population will converge, and the optimal solution of the objective function will be output. $v_{id}$ represents the current state search speed of the individual particle, $x_{id}$ represents the current search position of the individual particle, $P_{best}$ represents the optimal search position of the individual particle, and $G_{best}$ represents the global optimal position of the individual particle in the search domain. The population particle adjusts its own search direction and speed by cyclically tracking $P_{best}$ and $G_{best}$ in the population, and continuously optimizes the particle swarm search algorithm until the system records the individual extreme value and the overall extreme value to meet the predetermined value or the number of operation overflows. In the process of finding the optimal monomer extreme point, we can express the iterative process of particle velocity and position as Formulas (24) and (25).
(24)$$v_{i.d}^{\left(k+1\right)}=\omega v_{{}_{i.d}}^{\left(k\right)}+c_{1}r_{1}(P_{i.d}-x_{i.d}^{\left(k\right)})+c_{2}r_{2}(P_{g.d}-x_{g.d}^{\left(k\right)})$$
(25)$$x_{i.d}^{\left(k+1\right)}=x_{i.d}^{\left(k\right)}+v_{i.d}^{\left(k+1\right)}$$
where k is the iteration number, d is the dimension of the independent variable, and ω is the inertia weight, which is used to adjust the influence of the upper-dimensional search speed on the current speed. $r_{1}$ and $r_{2}$ are random numbers with a value range of [0,1], and $r_{1}$ and $r_{2}$ are independent of each other. In the variables, $x_{i}(k)$ is the position vector of particle *i* at the *k*-th iteration, and $v_{i}(k)$ is the velocity vector of the particle at the k-th iteration. $c_{1}$ is the acceleration weight coefficient of the detected particle itself, and $c_{2}$ is the weight coefficient of the global acceleration, which is used to adjust the movement speed of the particle at the optimal position of the individual and the overall optimal position. Generally, the coefficient is a constant between [0,2]. This coefficient can effectively improve the direction and trajectory algorithms of particles in the process of searching for optimal positions. The schematic diagram of particle search displacement is shown in Figure 6.

In the iterative formula of the PSO algorithm, Formulas (24) and (25) are composed of three parts: The particle is affected by the previous search speed $v_{id}(k)$ and its influence degree is represented by the inertia weight, the distance $P_{best}-x_{id}^{\left(k\right)}$ between the position of the particle *i* and the best position of the monomer sampling, and distance $G_{best}-x_{id}^{\left(k\right)}$ between the position of particle *i* and the optimal sampling position of the group. By introducing the ability of the self-searching strategy and selecting appropriate learning weights, the convergence speed of the algorithm can be greatly accelerated, and the situation of search oscillation caused by particles falling into local optimum during the search process can be avoided.

The PSO algorithm is an optimization algorithm established by simulating swarm intelligence, which breaks through the continuity and differentiability required in the mathematical model of the traditional signal system. Compared with other genetic algorithms, the PSO algorithm can save the optimal solutions of all particles before the current state, and will not be destroyed with the change in the population, which makes the algorithm have better stability. In addition, the PSO algorithm does not need coding, and the individual particles are updated through the internal velocity, the principle is simpler, and the required parameters are fewer. However, when the PSO algorithm is aimed at discrete problems, it still lacks a relatively mature convergence analysis method and needs to be further explored. In order to improve iterative efficiency, a lot of logic resources are often required, so it is necessary to make strict judgments on the number of particles when high real-time performance is required.

### 3.4. Particle Swarm Adaptive Interval Division

The sampling data function of the output signal of the photoelectric encoder is a nonlinear function. In the constrained optimization problem, in the early stage of the particle swarm algorithm execution, attention should be paid to the global particle search to maximize the exploration of the feasible area. In the later stage of the algorithm, local search needs to be strengthened to improve the algorithm’s finding of the optimal solution. In this paper, in order to better guide the individual population and promote the optimization process, through the adaptive interval division, the parallel particle swarm iteration algorithm is performed in the sub-region, and the local optimal value is found and stored in the external memory for optimal value comparison. As the number of particles in the external memory increases, the adaptive interval is further subdivided, which is beneficial to guide particles for further exploration in low-density areas or blank areas. This method is conducive to the development of feasible areas and avoids local convergence while maintaining population diversity.

The interval division method adopts the hexagonal area division of the bionic honeycomb structure, and the area division is determined by the distribution number of the area containing the optimal particles of the area in the current external memory. In the initial stage, the number of locally optimal individual particles in the external memory is small, and a looser particle area division method is adopted to avoid the appearance of particle-free areas. When the optimal particle region is found, the next step can be divided. No matter how many times the sub-region is divided, the sub-region can maintain the same search area, which effectively avoids the problem of individual particle optimization falling into the “local trap”. The target space division area is continuously improved with the increase in the number of optimal particles in the area in the external memory. This type of adaptive angle area division can gradually refine the area division and more effectively guide the population to explore areas with different particle densities, thereby increasing the probability of particle generation in extremely small areas.

Within the population search domain, the search interval should be adaptively divided according to the sub-area search density, and a single sub-area should be further divided into areas. In this paper, the area division is based on the honeycomb hexagonal structure. *D* represents the number of sub-areas included in the current area division state, and *i* represents the number of iterations of the area division in the current state. The upper limit of the external memory particle is 50 as an example.

Divide the population search space into $D_{0}=6$ regions, where *i* is the number of region divisions. As shown in Figure 7a, when the search area is divided for the first time, six effective search areas are obtained, and the search range of each group of areas is completely equal. As the number of algorithm iterations increases, the number of individual historical optimal particles extracted to the external memory increases. When the termination condition of the first division iteration is reached, the second region division is performed. In this paper, the number of particles in the external memory is represented by $P_{gather}$, and the external memory is essentially a comparator, which compares the extracted particles in the current optimal position of the individual to obtain the optimal position $G_{best}$ in the current population history. The optimal location acquisition is processed in parallel with the individual optimal location search in the subspace of the search domain.As shown in Figure 7b, as the number of iterations in the search domain is accumulated, the number of particles $P_{best}$ in the individual optimal historical position in the external comparator increases, and the number of particles in the search subspace also increases accordingly, so it is necessary to further subdivide the search area. Starting from i=2 iterations, the region adaptive division is performed:

(26)$$(i-1)N/i_{\max}<P_{gather}<iN/i_{\max}$$*N* is the maximum number of individual current optimal values set in the internal and external memory of the current iteration number, and $i_{\max}\leq \log_{D}^{N}$.

3.By analogy, the interval division of the population search domain can be continuously realized. Ideally, it should be ensured that there is one particle in each sub-region. However, due to the fast convergence speed of the PSO algorithm, even if the number of iterations is increased in some areas, better optimization results cannot be achieved. Therefore, according to the feedback state of the particle search space, some search areas without particles can be excluded accordingly, as shown in Figure 7c. In order not to affect the overall convergence speed of the algorithm, it is necessary to set the termination condition of the algorithm iteration. In this paper, we set the algorithm to stop when the algorithm reaches the predetermined maximum number of iterations or when the number of particles in the subspace of the search domain is 1.

### 3.5. Particle Swarm Multi-Swarm Parallel Iteration

During the execution of the standard particle swarm optimization, the system network structure is usually a single-community fully connected grid. The fully connected topology has better information transfer speed and algorithm convergence speed, and performs better in the search process of lower dimensions. However, in the higher-dimensional global optimal value optimization problem, the network structure model will be trapped in the optimal structure in some dimensions, resulting in a local optimal situation in the overall search of the population. Relatively speaking, the von Neumann topology, because of its three-dimensional network particle distribution structure, strengthens the exchange of particle information between multiple search areas and enables the population to have a stronger ability to get rid of local traps. This further strengthens the optimization ability of the algorithm, improves the search efficiency and accuracy of the algorithm, and avoids the algorithm falling into the local optimal situation. Based on the self-adaptive division of the search domain mentioned above, a composite mesh model combining fully connected topology and von Neumann topology with multi-group parallel cooperation is proposed. Different topology models use different weights to achieve better cooperation. The schematic diagram of the parallel topology algorithm is shown in Figure 8.

Combined with the adaptive region division proposed in Section 3.4, a fully connected topology is used in the divided sub-regions to explore the regional optimal value of a small number of particles, and the regional optimal value $P_{best}$ will be extracted to the external memory. The global exploration area adopts the von Neumann topology model. For each evolution, the particle extracts the global optimal value $G_{best}$ in the current iteration state and sends it to the fully connected topology of the sub-area in the form of information exchange. After the fully connected topology receives the $G_{best}$ sent by the von Neumann topology, it compares the optimal value $P_{best}$ obtained by the fully connected topology in the sub-region with the global optimal value $G_{best}$ obtained by the von Neumann topology model. If the regional optimal value $P_{best}$ is better than the global optimal value, the iterative information is fed back to the von Neumann structure to adjust its evolution rules. If the regional optimal value is worse than the global optimal value, the sub-regional evolution direction is adjusted according to the global optimal value.

In the particle search process, in order to effectively improve the algorithm’s search ability, it can effectively avoid falling into the “local optimal trap” when dealing with complex iterative problems. Starting from the structural characteristics of different topology types in the iterative process, the von Neumann topology is used as the global topology structure, and the fully connected topology is used as the sub-region population to explore the structure. Taking advantage of the information-sharing characteristics among multi-swarm particles and based on particle fitness, a particle swarm optimization algorithm for multi-structure parallel search is established. The two topology algorithms perform a parallel search for the global and sub-search domains of the population search domain. The particle swarm algorithm compares the local optimum and the global optimum through external memory and uses the comparison results to adjust the iterative strategy. Information sharing between topological structures adopts a secondary communication strategy. After each iteration, the subgroup and the population as a whole share information. At the beginning of each iteration, different population algorithms will detect whether the target population needs to update the evolutionary rules to ensure that the population evolves in the optimal direction and avoids falling into the trap of local optimality. The flowchart of the parallel iterative particle swarm algorithm is shown in Figure 9.

## 4. FPGA Implementation of Particle Swarm Optimization Algorithm

### 4.1. FPGA Logic Design of Particle Swarm Optimization Algorithm

The particle swarm algorithm requires high resource weighting and computing speed. FPGA is the most mainstream hardware circuit-bearing platform at this stage. The computing method is the parallel computing mode of the pipeline architecture. This method can greatly improve the computing speed and meet the system performance requirements. In the process of building the FPGA platform for the particle swarm algorithm, it mainly includes the following four modules: a digital signal acquisition module, particle velocity update module, particle position update module, and data processing module.

(1)Digital Signal Acquisition Module

The digital signal acquisition module adopts an AD9248 digital acquisition chip. The chip adopts a multi-stage differential pipeline architecture, has built-in output error correction logic, and can provide 16-bit accuracy at the highest data rate of 65 MSPS. A double single-ended independent clock input is used to control the internal conversion cycle. In the double-frame acquisition mode, the data are output through two pins—$D_{1}$ and $D_{0}$—of which the $D_{1}$ pin outputs the high byte and the $D_{0}$ pin outputs the low byte. The deserialization operation of the sampled data is completed through the FPGA internal IP core LVDS_RX. The deserialization clock is set to 50 MHz in this paper, and the data bus bandwidth is designed to be 640 Mbps. The 16-bit test data source is generated according to the actual sampling data, in which the high-order 8-bit data decrease in turn, and the low-order 8-bit data increase in turn. The simulation results in Modelsim are shown in Figure 10.

(2)Particle Velocity Update Module

The particle swarm iteration speed update thinking expressed by Equation (24) is set, the particle swarm speed update module is set, and the inertia weight is set to a fixed value of 0.8. The particle search speed is updated according to the particle speed stored in RAM, as well as the global optimal value and local optimal value appearing in the historical search process. In the design process of the speed update module, a judge is added to prevent the particles from exceeding the predetermined search boundary due to the high search speed during the particle speed update process, resulting in repeated oscillations at the search boundary. v is the current particle search speed, $c_{1},c_{2}$ is the acceleration factor, $r_{1},r_{2}$ is a random number, pop is the particle, and the output value update is the particle update speed. The particle velocity update FPGA module is shown in Figure 11.

(3)Particle Position Update Module

From Formula (25), the particle position update method in the PSO algorithm can be obtained. In the design of the FPGA logic module, a judger is also added to ensure that the particle search process is limited within the search domain. The testing process mainly includes a serial–parallel conversion module, data ROM module, weight–threshold processing module, weight–threshold conversion module, output layer module, subtraction module, and so on. The weight–threshold processing module can split the particles according to a predetermined grid structure. Assuming that the network has n node input layers, m node hidden layers, and l node output layers, the algorithm particle dimension can be expressed as (n+1)×m+(m+1)×l. Convert the weights and thresholds generated by the weight–threshold processing module into a structure suitable for FPGA logic calculation. In order to facilitate parallel computing, the weights and thresholds generated by the weight–threshold processing module are converted into parallel output data. The input of the serial–parallel conversion module is the global optimal individual after the iteration is completed, and the subtraction module calculates the difference between the test data and the simulation data. The particle position update FPGA module is shown in Figure 12.

### 4.2. FPGA Key Parameter Setting

When the PSO algorithm is used to compensate the subdivision error of the photoelectric encoder, a larger initial population size can better realize the global exploration ability of the particle swarm to the target interval, whereas a smaller initial population can complete the rapid convergence of the algorithm. Therefore, it is necessary to combine the stability and effectiveness of the algorithm to select appropriate population parameters so that the error compensation algorithm can achieve better performance. In addition, since the particle swarm optimization algorithm has high requirements for the computing speed and resources of the hardware platform in the compensation process, it is necessary to select appropriate algorithm parameters. Therefore, in Section 4.2, we set the inertia weight mode and the number of particles contained in the particle swarm in the PSO algorithm based on the application scenario of the PSO algorithm and the occupancy of the logic resources of the FPGA platform.

(1)Inertia Weight

In the process of local area and global area search, inertia weight plays a very important role. Usually, there are two change modes of linear inertia weight and nonlinear inertia weight. For the moiré grating output signal, in the variation mode of nonlinear inertia weight, the fitness value of the PSO algorithm fitting degree test result fluctuates less and is relatively stable. The linear inertia weight change mode is less stable than the nonlinear weight change mode, but from the perspective of fitting accuracy, the linear weight change mode is more accurate at the optimal point. The fitness analysis of linear inertia weight and nonlinear inertia weight is shown in Figure 13.

From the analysis of the logical occupancy, the same population quantity and spatial dimension distribution are selected to compare the logical resource occupancy of the two inertia weight changes. It can be seen from the actual simulation results that the number of logic resources occupied by the linear inertia weight change method in the FPGA is slightly lower than that of the nonlinear inertia weight logic resource occupancy.

In general, a larger inertia weight can help particles effectively avoid falling into a local optimum during the search process and improve the algorithm’s search ability, whereas a smaller inertia weight can help improve the particle’s convergence ability during the global search process. The inertia weight should also be dynamically adjusted with the performance of the current optimal value of the population and the number of algorithm iterations. In the study by [20], a linear inertia weight change Formula (31) is proposed:(27)$$\omega =\omega_{\max}-\frac{(\omega_{\max}-\omega_{\min})\cdot k}{k_{\max}}$$

Among them, $\omega_{\max}$ is the maximum value of the inertia weight, $k_{\max}$ is the upper limit of the set number of iterations, and k is the current number of iterations. In this paper, the value variation range of the inertia weight is ω∈[0.4,1].

(2)Number of Particles

The size of the number of particles will affect the convergence and accuracy of the algorithm to a certain extent. In general, optimization algorithms in different scenarios will choose different particle numbers. For most problems, N∈[10,50]. A smaller number of particles can make the particle swarm algorithm achieve a better population convergence effect, but when the population search domain is large or in a more complex optimization problem, a larger number of particles should be selected to meet the search effect. In this paper, in the problem of setting the number of particles in the population, the FPGA platform is used to test the sampling signal particles of grating moiré fringes. By controlling other variables to be exactly the same, the comparison of the system logic resource occupancy under the condition of different population particle numbers is achieved. The test results are shown in Table 1.

It can be concluded that when the other variables are exactly the same, as the number of particles in the population increases, the algorithm improves the search efficiency of sub-regions and the fluctuation of the optimal fitness value during the search process is relatively reduced. From the data analysis and comprehensive analysis of the FPGA logic resource occupancy, the number of particles is positively correlated with the FPGA logic resource occupancy. However, when the number of particles is 20 and the number of particles is 30, the difference between the optimal fitting accuracy values is small, but the use of 30 search particles takes up more logic resources, so this paper chooses N=20 as the number of particles in the test population.

In order to test the performance superiority of the FPGA platform, under the same experimental conditions, FPGA and MATLAB are used to realize the fitness of the PSO algorithm to de-encode sampling data function optimization. The test results are shown in Figure 14.

From Figure 14, it can be concluded that FPGA can better complete the population convergence, and the overall performance is higher than that of MATLAB and other system execution platforms.

## 5. Experimental Design

In this paper, in order to verify the effectiveness of the subdivision error compensation algorithm proposed above, the experimental signal is loaded into the precision error detection and compensation platform written with the algorithm execution program to verify the function of the algorithm. The data acquisition and calculation development board of the test platform is shown in Figure 14. The AD acquisition module adopts an AD9248 conversion board. The FPGA development board selects ALTERA’s DE2-115 to implement the error compensation algorithm. The algorithm-writing work is carried out in a Quartus II development environment using Verilog HDL language to design the compensation algorithm circuit and download it to the FPGA chip to complete the compensation task.

In order to cooperate with the verification test results, a photoelectric encoder error detection platform is designed as a special error detection and compensation device. The internal structure of the subdivision error detection and compensation platform is shown in Figure 15. The platform uses an autocollimator and a regular tetrahedron to form an optical position closed loop as an angle reference, and a fixed-position optical reflection device is also installed to detect the signal drift of the autocollimator during the process. The detection device can perform ±360° continuous or fixed compensation detection, and adopts an internal and external dual-axis structure. The positioning accuracy of the angular position of the inner and outer shafts is ±0.2″, the angular position stability is ±0.36″, the verticality of the shaft system and the axis rotation accuracy error are ≤±2″, and the angular position resolution is ≤0.36″. The detection device consists of a detection platform and a control box that can transmit the collected data to the host computer in real time for display. The size of the inner table at the top of the detection device is designed to be Φ210 mm, and the material is super-hard aluminum alloy. The flatness of 0.005 mm and the runout of the end face of 0.01 mm are guaranteed by grinding the table surface and the installation positioning surface. Load-mounting screw holes are arranged on the table, and by matching the clamping flanges with corresponding dimensions, the detection of encoders of any size with a diameter of less than 210 mm and a single weight of less than 2.5 kg can be realized. The structure designed in this paper can form an optical closed-loop detection structure and has a relatively low cost to complete higher-precision detection, which effectively improves the detection efficiency and accuracy. The system structure of the subdivision error detection and compensation platform is shown in Figure 16.

In order to verify the experimental effect of the algorithm proposed in this paper, we compare the signal subdivision results of the output signal of the photoelectric encoder before and after compensation under the same experimental environment. As shown in Figure 17, the subdivision result in one sinusoidal signal period outputted by the detection platform is obtained, and the unit of the curve is the subdivision value.

From the image analysis, it can be concluded that in the uncompensated subdivision period, using the normal subdivision look-up table, the deviation value of the signal angle subdivision point connection reaches the peak value at *π*/2 and 3*π*/2. The subdivision value corresponding to the *π*/2 subdivision point is 1.81 radians, and the subdivision value corresponding to the 3*π*/2 subdivision point is 4.89 radians. When the optimized subdivision look-up table is used to compensate the detection error, it can be seen from the blue marked points in Figure 17 that the connection of the angle subdivision points of the signal is close to the line of subdivision points in the ideal state at *π*/2 and 3*π*/2—that is, at the peak of the error, the orthogonality deviation is significantly reduced. The subdivision value corresponding to the *π*/2 subdivision point is 1.51 radians, and the subdivision value corresponding to the 3*π*/2 subdivision point is 4.54 radians.

In order to verify the effect of error compensation, 64 points are measured in one subdivision period selected by the specific angular position of the 25-bit encoder. The subdivision accuracy before and after the compensation of the photoelectric signal of the encoder is detected, and the detection result of the dynamic subdivision error is shown in Figure 18:

As can be seen from Figure 16, the maximum value of the subdivision error before compensation is +0.50″, the minimum value is −0.49″, and the peak-to-peak error is 0.99″; after the algorithm compensation, the subdivision error value is significantly reduced, the maximum error value is +0.31″, the minimum error value is −0.16″, the peak-to-peak error is 0.47″, and the subdivision accuracy is significantly improved. This algorithm is effective for quadrature errors. The dynamic subdivision error after compensation is about the same as that before compensation 12.

It can be concluded from Figure 19 that the maximum value of the subdivision error before compensation is +0.48″, the minimum value is −0.78″, and the peak-to-peak error is 1.26″; after the algorithm compensation, the subdivision error value is significantly reduced, the maximum error value is +0.16″, the minimum error value is −0.31″, the peak-to-peak error is 0.48″, and the subdivision accuracy is significantly improved. In addition, a small numbers of systematic errors and random errors of readings are still included in the detection of subdivision errors using the precision error precision-measuring platform.

Fourier spectrum analysis is performed on the above subdivision error compensation results, and the results are shown in Figure 20. Before subdivision error compensation, the influence of low-order signals (first-order, second-order, third-order, and fourth-order signals) in the subdivision error spectrum is more obvious; after the subdivision error compensation operation, the low-order interference components are effectively suppressed, and the overall high-order harmonic components of the subdivision error of the signal are significantly reduced. From the analysis of the subdivision error results in the time and frequency domains, it can be seen that the moiré grating signal subdivision error compensation scheme based on the particle swarm optimization algorithm proposed in this paper is effective and feasible, and the subdivision accuracy is improved significantly.

## 6. Conclusions

To sum up, the subdivision error compensation method of the moiré grating fringe signal is studied in this paper. According to the characteristics of the grating signal waveform equation, the generation principle of grating moiré fringe subdivision error is analyzed by a mathematical model. A search domain subdivision method based on hive-adaptive area division is proposed, and a multi-swarm optimization search algorithm using parallel iteration is proposed by taking advantage of the pipeline architecture of the FPGA development platform and the advantages of parallel processing. Using the advantages of the fully connected topology and the von Neumann topology, the rapid iterative convergence of the particle population search domain is realized, which effectively improves the convergence speed and accuracy of the traditional PSO algorithm. Based on the FPGA development platform and the optical detection device, a photoelectric encoder moiré grating subdivision signal error detection and compensation device is constructed, and the dynamic/static error compensation test of the 25-bit photoelectric encoder is carried out. The test results show that the optimized error compensation algorithm reduces the dynamic subdivision error of the system from the original 0.9928″ to 0.4779″, the static subdivision error is reduced from 1.2641″ to 0.4866″, and the overall system can obtain better convergence efficiency and a more accurate fitness value.

## Figures and Tables

**Figure 1 sensors-22-04456-f001:**
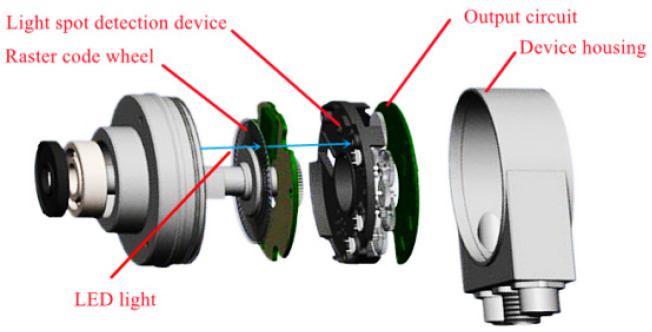
Photoelectric encoder explosion structure diagram.

**Figure 2 sensors-22-04456-f002:**
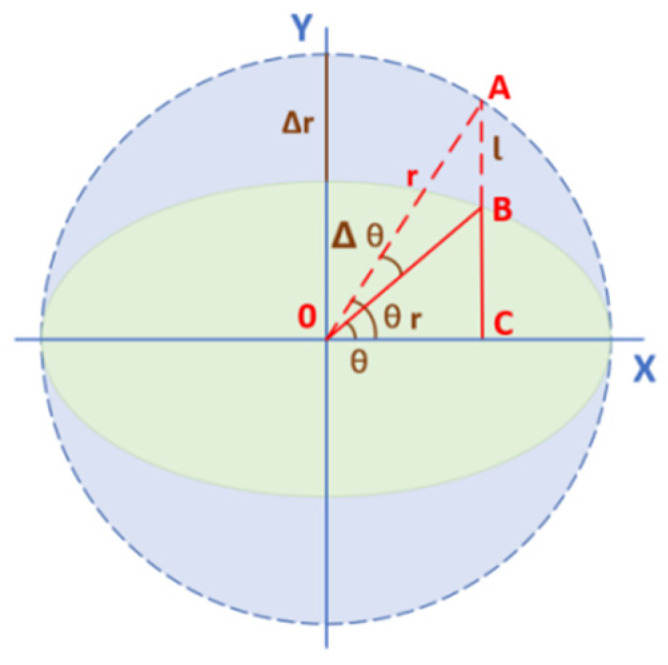
Graphical analysis of the Lissajous sine error.

**Figure 3 sensors-22-04456-f003:**
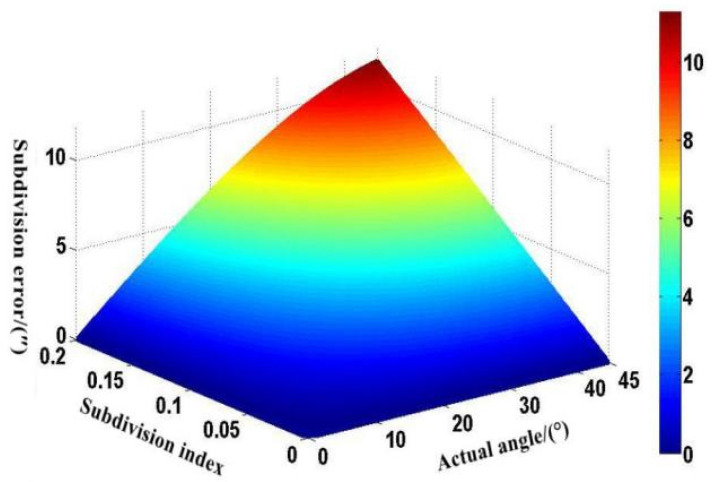
Subdivision error introduced by signal sinusoidal error [19].

**Figure 4 sensors-22-04456-f004:**
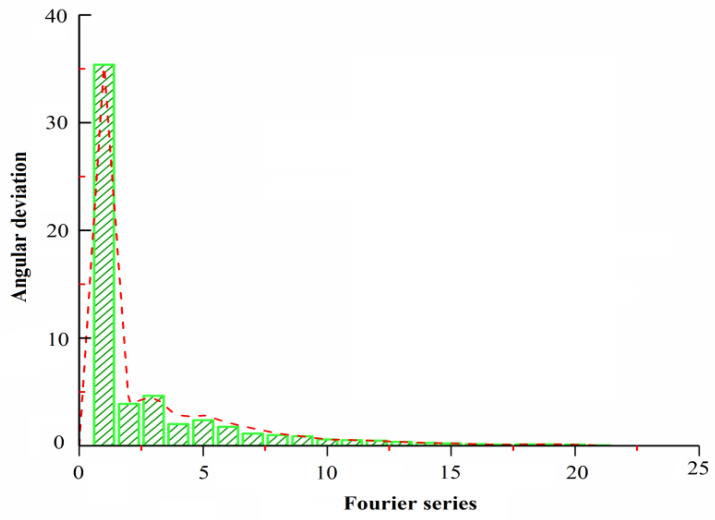
Analysis of high-order harmonic error.

**Figure 5 sensors-22-04456-f005:**
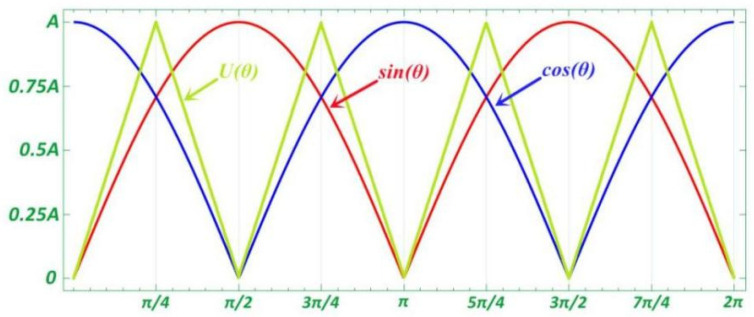
Comparison among original signals and constructed function.

**Figure 6 sensors-22-04456-f006:**
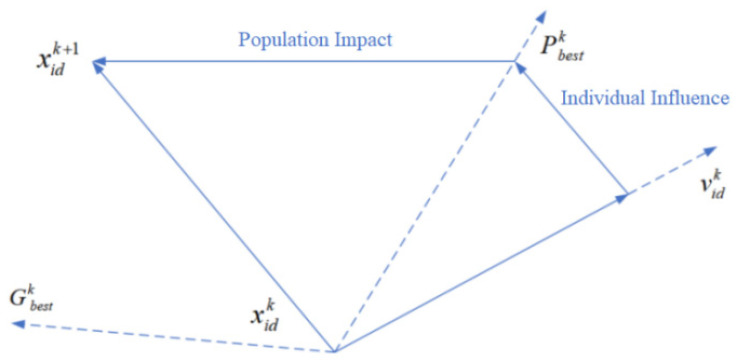
Schematic diagram of particle displacement.

**Figure 7 sensors-22-04456-f007:**
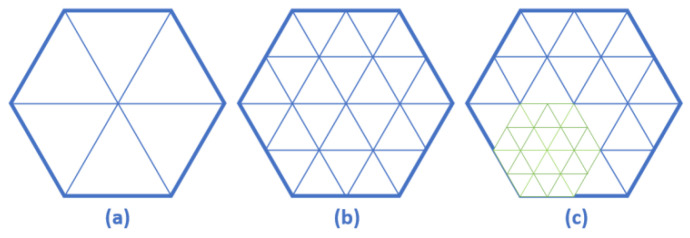
Adaptive sampling area division of the honeycomb structure (*D* = 6). (**a**) Partition of particle initial iterative region; (**b**) 2 iterations to search the domain model; (**c**) Schematic representation of iterative derivation of the search domain.

**Figure 8 sensors-22-04456-f008:**
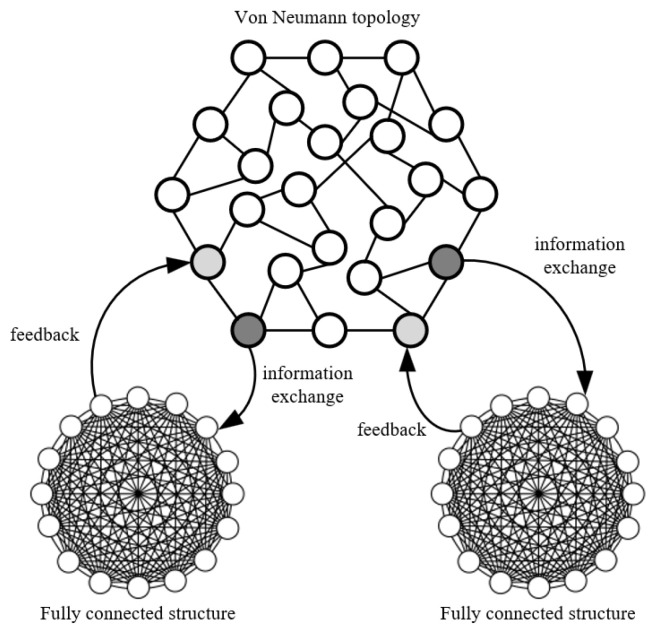
Schematic diagram of the parallel topology algorithm.

**Figure 9 sensors-22-04456-f009:**
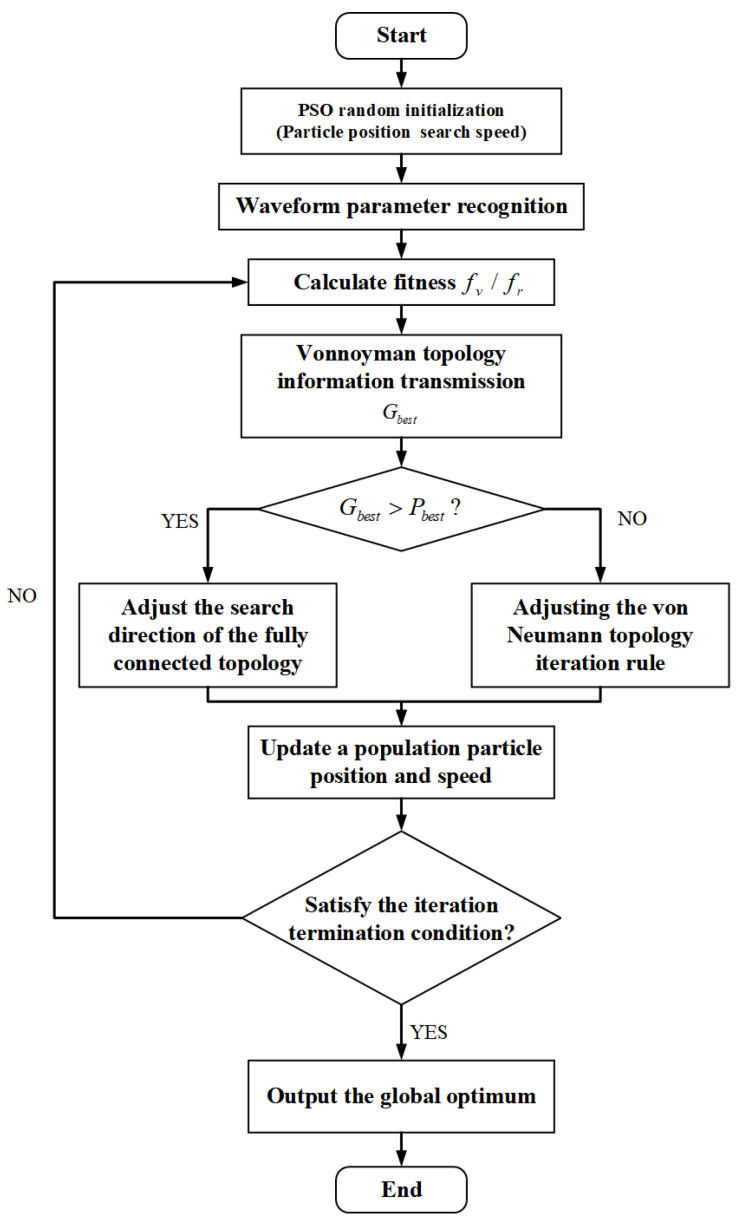
Flow chart of the parallel iterative particle swarm algorithm.

**Figure 10 sensors-22-04456-f010:**
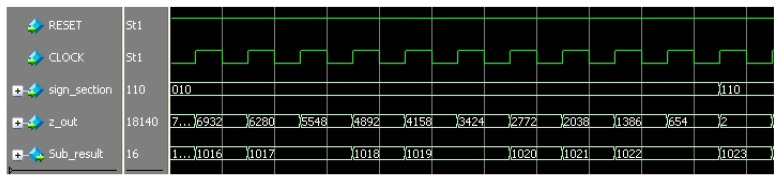
Subdivision circuit simulation test results.

**Figure 11 sensors-22-04456-f011:**
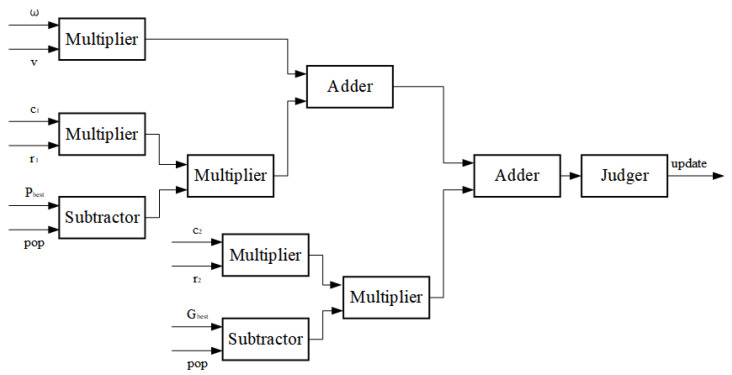
FPGA particle velocity update module.

**Figure 12 sensors-22-04456-f012:**
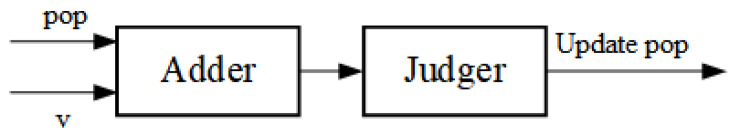
Particle position update module.

**Figure 13 sensors-22-04456-f013:**
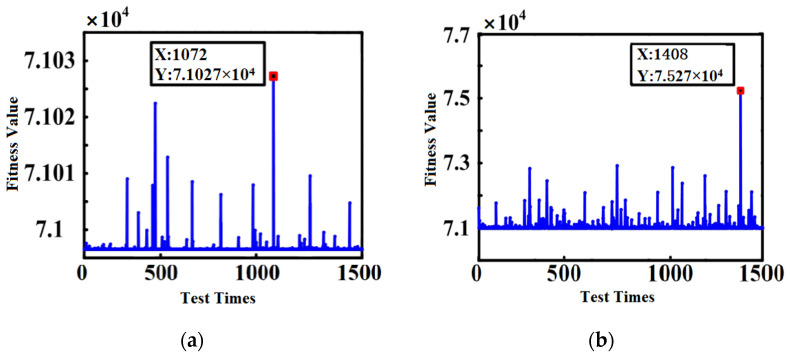
Comparison of fitness of linear and nonlinear inertial weight transformation methods. (**a**) Linear inertia weight change method; (**b**) non-linear inertia weight change method.

**Figure 14 sensors-22-04456-f014:**
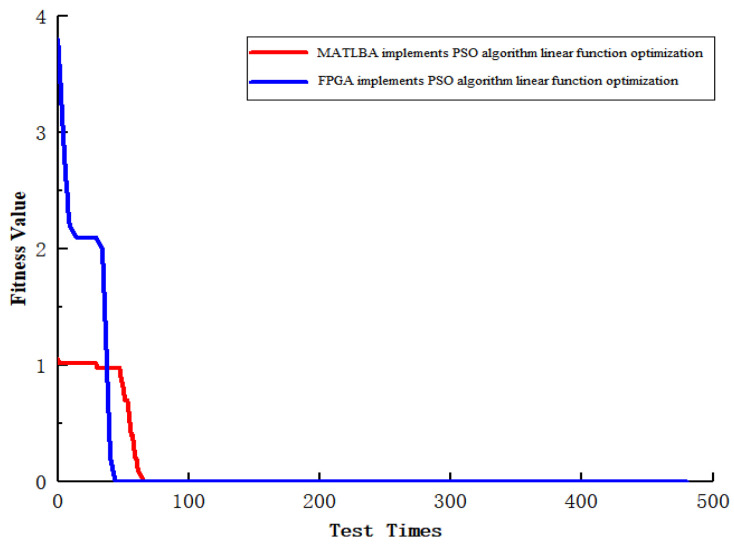
The fitness diagram of the PSO algorithm implemented by FPGA and MATLAB.

**Figure 15 sensors-22-04456-f015:**
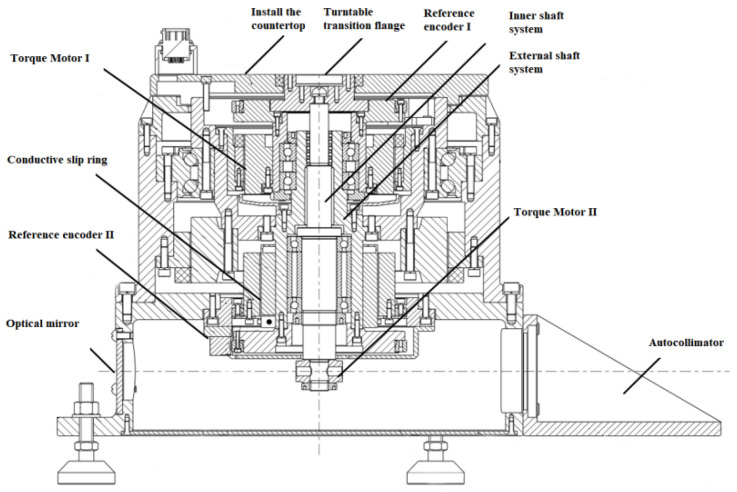
Moiré grating interpolation error precision detection platform.

**Figure 16 sensors-22-04456-f016:**
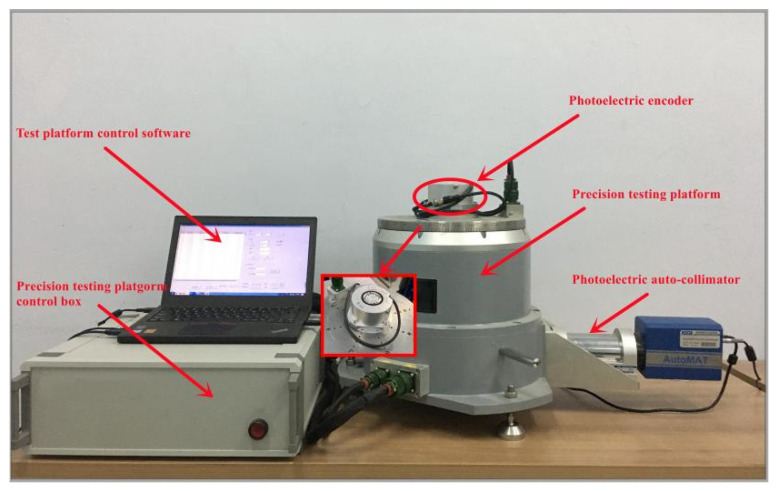
Moiré grating interpolation error precision detection platform.

**Figure 17 sensors-22-04456-f017:**
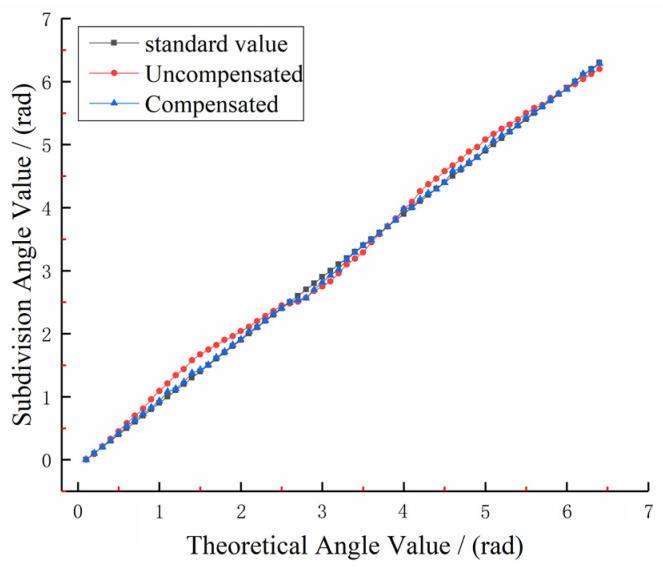
Subdivision error of the signal before and after error compensation.

**Figure 18 sensors-22-04456-f018:**
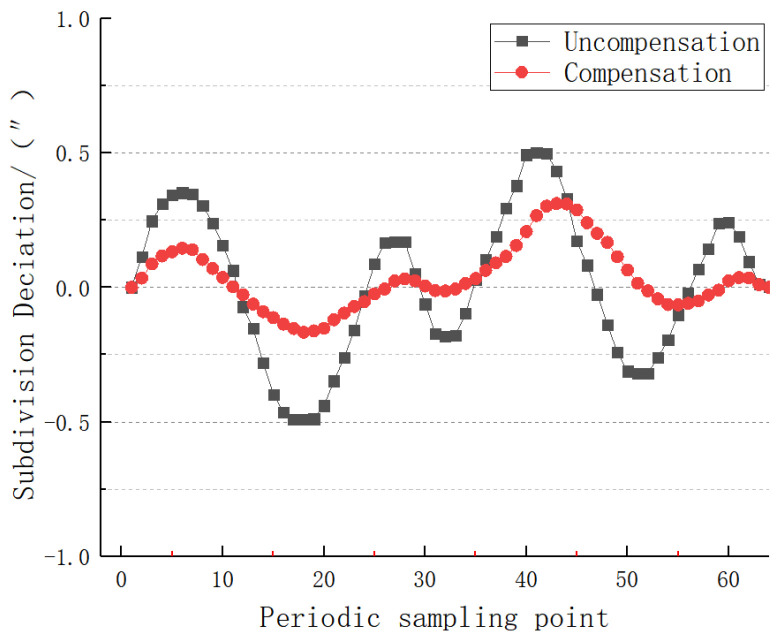
Comparison before and after dynamic subdivision error compensation.

**Figure 19 sensors-22-04456-f019:**
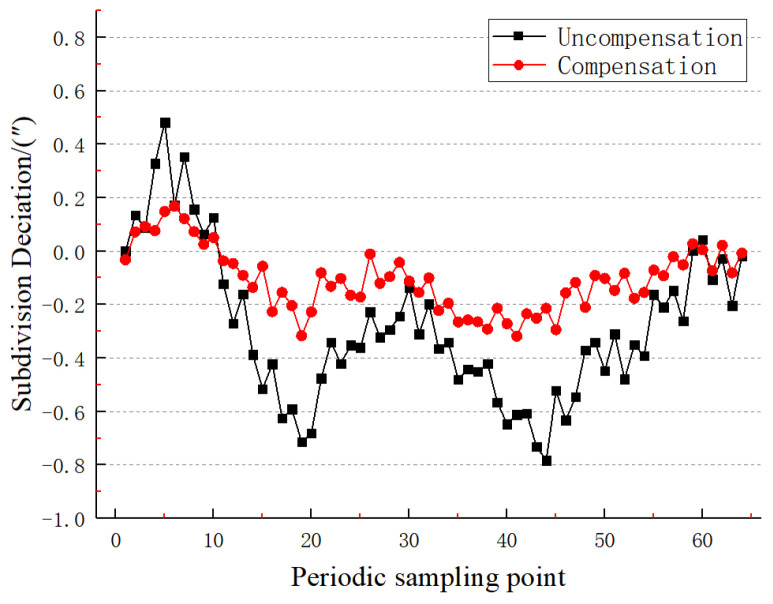
Comparison before and after static subdivision error compensation.

**Figure 20 sensors-22-04456-f020:**
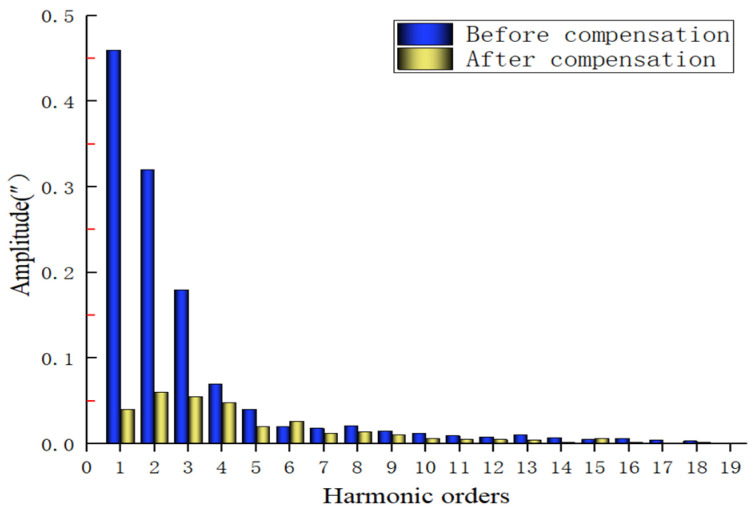
Spectrum results of the subdivision error before and after error compensation.

**Table 1 sensors-22-04456-t001:** FPGA resource occupation under different inertia weight methods and particle numbers.

Particle Population	Space Dimensionality	Changing Mode	LE	Best Fitness
20	5	Nonlinear inertia weight	13,852	
10	5	Linear inertia weight	7697	2.638 × 10^5^
20	5		13,067	7.41 × 10^4^
30	5		18,673	6.97 × 10^4^

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
