# Peer review of "Research on Particle Swarm Compensation Method for Subdivision Error Optimization of Photoelectric Encoder Based on Parallel Iteration"

_sensors, 2022, doi:10.3390/s22124456_

Round 1
Reviewer 1 Report
The presented manuscript is not clearly written and it is needed some substantial revisions.

Reviewer 2 Report
Dear Authors,
The presented manuscript describes a method, which allows increasing of an angular positioning sensor precision. The novelty consists of the use of global optimization technique to reduce the sub-division error. Thus, the topic and content of the manuscript match the Sensors journal scope.
However, I advice to pay attention to the following remarks prior to acceptance of the paper:
MAJOR POINTS:
1. There is no general description of the sensor and its' operation principle with a schematic. It would be useful to have it. Also, In the beginning of sec 3.1 it is mentioned that the grating diffracts the light, while normally a Moire-effect sensor uses just amplitude modulation.
2. In general, the paper is huge and not that well organized. It contains too many of vague text fragments ith multiple self-repeats. Some parts like lines 632-636 or 700-719 could be shortened or replaced by tables. In general, it is really difficult to read. I wouldn't expect this from a paper on a relatively narrow topic.
3. There are multiple sefl-repeats in equations - see Eq 12 and 16, 3 and 21 etc. Also, some equations like 22,23,25 seem to be too self-evident.
4. Sec 4.1 - What are the alternative hardware/software solutions? Is it worth to build such a copmlicated system for a single sensor?
5. What is used as a reference for the experimental measurements? And how to separate the sources of error in the experimental data?
MINOR POINTS:
1. Use of acronyms is not always correct. Normally they are introduced once with a full spelling and then used throughout the text. In the Abstract and sec.1 it happens in the opposite way.
2. Numerical precision - the angular errors are given with a 3-4 digits precision. I'm afraid that 0.0001" is a purely computational matter with no physical meaning.
3. Seems like some references are missing between lines 57-60.
4. Is Fig.3. referenced in the text?
5. Line 260 - should it be infinitely close or just close to 0?
6. Sec 3.3. - I guess there should be references to the initial PSO method and its' use in similar tasks.
7. Line 320 - perhaps, it's "merit function", not "cost function"
8. Fig.6. and further - it's not clear ho to deal with different number of variables with this interpretation of the search method.
9. Fig.13 - which method implemented in MATLAB is referred to exactly ?
10. Table 1 - is this LE difference really significant?
11. Lines 673-678 and Fig.14,15 - the menthioned optical hardware is not seen on the figures, while the photos are not very informative. There is no measurement priniple in the beginning to compare with, there are no labels indicating the parts or arrows to show the beam or dataflow. So, these figures are barely useful.
To summarize, I think that the manuscript in it's present state would be of a relatively low interest for the community. It should be shortened and re-written with a much clearer structure and description of the experimental work.
Round 2
Reviewer 1 Report
The improved form of the submitted manuscript merits the publication.
Author Response
No more suggestions from reviewers
Reviewer 2 Report
Dear Authors,
Thank you for taking my comments into account when preparing the revised version of the manuscript. There are some minor comments related to the newly added parts of the manuscript:
1. Fig .1 - I would expect to see an explicit geometrical definition of the parameters you are operating with afterwards.
2. Section 3.3. - I'm still missing a reference to the original work introducing the PSO method with all the corresponding math. The new paragraph is quite general and adds little for the method understanding.
3. Fig. 14-15 are more informative now. Apparently, the encoder itself is small in comparison to the entire setup. What are the exact dimensions? Also, is it correct that the reference platform provides a precision of 0.36″? Then how one can judge about ~0.0001" deviation?
In general, I think that the research topic is relatively narrow, but the approach used here is of a certain novelty and the claims are backed by experimental results, so the paper can be accepted for publication after fixing a few minor points.
